# Significance of Immunosuppressive Cells as a Target for Immunotherapies in Melanoma and Non-Melanoma Skin Cancers

**DOI:** 10.3390/biom10081087

**Published:** 2020-07-22

**Authors:** Taku Fujimura, Setsuya Aiba

**Affiliations:** Department of Dermatology, Tohoku University Graduate School of Medicine, 1-1 Seiryo-machi, Aoba-ku, Sendai 980-8574, Japan; saiba@med.tohoku.ac.jp

**Keywords:** TAMs, MDSCs, Tregs, TANs, soluble CD163, chemokines, PD1/PD-L1 signaling, anti-PD1 Abs

## Abstract

Tumor-associated macrophages (TAMs) have been detected in most skin cancers. TAMs produce various chemokines and angiogenic factors that promote tumor development, along with other immunosuppressive cells such as myeloid-derived suppressor cells (MDSCs), regulatory T cells (Tregs) and tumor-associated neutrophils. TAMs generated from monocytes develop into functional, fully activated macrophages, and TAMs obtain various immunosuppressive functions to maintain the tumor microenvironment. Since TAMs express PD1 to maintain the immunosuppressive M2 phenotype by PD1/PD-L1 signaling from tumor cells, and the blockade of PD1/PD-L1 signaling by anti-PD1 antibodies (Abs) activate and re-polarize TAMs into immunoreactive M1 phenotypes, TAMs represent a potential target for anti-PD1 Abs. The main population of TAMs comprises CD163^+^ M2 macrophages, and CD163^+^ TAMs release soluble (s)CD163 and several proinflammatory chemokines (CXCL5, CXCL10, CCL19, etc.) as a result of TAM activation to induce an immunosuppressive tumor microenvironment together with other immunosuppressive cells. Since direct blockade of PD1/PD-L1 signaling between tumor cells and tumor-infiltrating T cells (both effector T cells and Tregs) is mandatory for inducing an anti-immune response by anti-PD1 Abs, anti-PD1 Abs need to reach the tumor microenvironment to induce anti-immune responses in the tumor-bearing host. Taken together, TAM-related factors could offer a biomarker for anti-PD1 Ab-based immunotherapy. Understanding the crosstalk between TAMs and immunosuppressive cells is important for optimizing PD1 Ab-based immunotherapy.

## 1. Introduction

Tumor-associated macrophages (TAMs) have been detected in most skin cancers [1]. TAMs produce various chemokines that attract other immunosuppressive cells such as myeloid-derived suppressor cells (MDSCs), regulatory T cells (Tregs) and tumor-associated neutrophils (TANs) to maintain an immunosuppressive tumor microenvironment [1]. TAMs also produce matrix metalloproteinases (MMPs), which play critical roles in the tissue remodeling associated with protein cleavage to modify the immune microenvironment, angiogenesis, tissue repair, local invasion, and metastasis [1,2]. In addition, TAMs express immune checkpoint modulators (e.g., programmed death ligand 1 [PD-L1], B7-H3, B7-H4) [3] that directly suppress activated T cells. Moreover, TAMs also express PD1, which is necessary for maintaining M2 phenotypes in TAMs via PD-L1/PD1 signaling from tumor cells [4]. Taken together, TAMs are a heterogeneous population of macrophages that play a central role in the induction of immune tolerance in the tumor microenvironment [1].

Not only TAMs, but also other immunosuppressive cells such as MDSCs, Tregs and TANs, should be taken into account when evaluating the immunosuppressive microenvironment of skin cancers [5,6,7]. Similar to TAMs, both MDSCs and TANs directly or indirectly suppress anti-tumor immune response [6,7], whereas Tregs directly suppress tumor-specific cytotoxic T cells in the tumor microenvironment [5]. Notably, environmental risk factors for skin cancer (e.g., sun exposure, chemical exposure) have been widely reported [8]. These risk factors modulate the profiles of tumor-infiltrating leukocytes (TILs), at least in part, through aryl hydrocarbon receptor (AhR)-dependent signal pathways [9]. Chronic exposure to AhR ligands at skin lesions is known to induce chronic inflammation, including macrophages, neutrophils and T cells [10]. Skin cancer is thus one of the optimal models to discuss the development of immunosuppressive microenvironments in cancers.

Since PD-L1/PD1 signaling is necessary for maintaining TAMs as immunosuppressive macrophages in PD-L1-expressing cancers such as melanoma, non-small cell lung cancer, colorectal cancer and Hodgkin’s lymphoma [4,11,12], anti-PD1 antibodies (Abs) such as nivolumab and pembrolizumab could activate and re-polarize TAMs into anti-tumor macrophages. In another report, Wang et al. reported that PD1^+^ TAMs suppress CD8^+^ T-cell function in gastric cancer [11]. More recently, Li et al. reported that exosomal HMGB1 could trigger the generation of PD1^+^ TAMs in esophageal carcinoma [12]. Notably, anti-PD1 Abs are useful and clinically permitted to be used for these cancer species. Taken together, anti-PD1 Abs could not only abrogate the immune suppression and re-activate CD8^+^ cytotoxic T cells [5], but also activate TAMs to induce an anti-tumor immune response by blocking of PD-L1/PD1 signaling pathway. Not only TAMs, but also MDSCs and Tregs help maintain an immunosuppressive microenvironment through PD-L1/PD1 signaling [3]. MDSCs can induce Tregs [13], and Tregs regulate the immunosuppressive function of MDSCs through PD-L1 [3].

This review discusses the differentiation, activation and immunosuppressive functions of TAMs, MDSCs, Tregs and TANs, and their benefits in cancer immunotherapy.

## 2. Significance of Immunosuppressive Cells in Developing Skin Cancers

### 2.1. Significance of TAMs in Developing Skin Cancers

#### 2.1.1. Chemokines from TAMs Determine Profiles of Tumor-Infiltrating Lymphocytes (TILs) in the Tumor Microenvironment

Since TAMs are stimulated by stromal factors, and produce characteristic chemokines in each tumor site in melanoma and non-melanoma skin cancers [1], understanding the correlations between chemokines derived from TAMs and stromal factors in each cancer species is important. The extracellular matrix protein periostin (POSTN) is expressed in the region surrounding melanoma cell nests in metastatic melanoma lesions [14], and could be a stimulator for TAMs in melanoma [1]. Notably, CD163^+^ M2 macrophages increase the production of chemokine C-C motif (CCL)17 and CCL22, both of which are known to recruit regulatory T cells (Tregs), by POSTN stimulation in vitro [15], and chemokine production is suppressed by type I interferons (IFNs) [16,17], suggesting that TAMs could also be used as a target of immunotherapy. Indeed, Georgoudaki et al. reported that TAMs derived from mouse B16 melanoma expressed macrophage receptor with collagenous structure (MARCO), and intravenous administration of anti-MARCO antibodies (Abs) reprogrammed the TAMs population to a proinflammatory phenotype and increased tumor immunogenicity [18]. In another report, IFN-β decreases the production of CCL22 from TAMs in B16F10 melanoma, leading to suppression of tumor growth by the modulation of TIL profiles in vivo [17]. Based on these pre-clinical findings of TAM-targeting therapies, a clinical study has already been undertaken [19]. Taken together, these reports suggest the significance of chemokines from TAMs that can be influenced by stromal factors to induce melanoma-specific profiles of TILs in melanoma.

Non-melanoma skin cancers such as extramammary Paget’s disease (EMPD), cutaneous squamous cell carcinoma (cSCC) and Merkel cell carcinoma (MCC) also possess heterogeneous CD163^+^ TAMs that could secrete an array of cytokines and chemokines in lesional skin to regulate the tumor microenvironment [1,20,21,22]. For example, serum sCD163 is increased in patients with EMPD compared to healthy donors [23], suggesting that CD163^+^ TAMs are constitutively activated in the lesional skin of EMPD. Indeed, soluble receptor activator of nuclear factor kappa-B ligand (RANKL) released by Paget’s cells activates TAMs and increases the production of CCL5, CCL17 and chemokine CXC motif (CXCL)10 from RANK^+^CD163^+^ M2 polarized TAMs [20]. These data suggested that sCD163 could represent a biomarker for the progression of EMPD. On the other hand, as Petterson et al. reported [21], CD163^+^ TAMs in cSCC heterogeneously polarized from M1 to M2, suggesting heterogeneous activation states of TAMs. CD163^+^ TAMs contribute to the tumor microenvironment in MCC to promote tumor development by inducing lymphangiogenesis and immunosuppressive cells such as Tregs [22,24]. These reports suggested that CD163^+^ TAMs could represent a therapeutic target for the treatment of these non-melanoma skin cancers.

#### 2.1.2. Angiogenic Factors from TAMs

TAMs produce angiogenic factors such as vascular endothelial growth factor (VEGF), platelet-derived growth factor, and matrix metalloproteinases (MMPs) to induce neovascularization [1,25,26,27]. Recent reports have suggested that melanoma-derived factors could differentiate M2 macrophages that produce angiogenic factor such as VEGF and MMP9 [25,27]. Among these, Tian et al. reported that expression of tripartite motif (TRIM)59 on TAMs attenuates the tumor-promoting effect of TAMs by inhibiting MMP9 expression on melanoma cells [25]. They conclude that TRIM59 in TAMs could be a potential regulator of tumor metastasis, and thus provide a target for immunotherapy [25]. Notably, MMP9 facilitates MMP9-dependent cleavage of PD-L1 surface expression, leading to anti-PD1 Ab resistance [27]. Taken together, the decreased expression of MMP9 achieved by targeting TAMs would suppress anti-PD1 Ab resistance by inhibiting PD-L1 downregulation.

Overall, TAMs produce a series of chemokines and angiogenetic factors under the stimulation of cancer-specific stromal factors to maintain an immunosuppressive tumor microenvironment in each cancer species.

### 2.2. Myeloid-Derived Suppressor Cells (MDSCs)

#### 2.2.1. Significance of MDSCs in Developing Skin Cancers

MDSCs are one of the key types of immunosuppressive cells with heterogeneous cell populations that can be found in tumor-bearing mice and in patients with cancer (Table 1) [6]. In humans, MDSCs are defined by a combination of several surface markers (e.g., CD11b^+^CD14^−^HLA-DR^−^ for monocytic (Mo-)MDSCs, or CD11b^+^CD14^−^CD33^+^CD15^+^CD66b^+^ for granulocytic (G)MDSCs) [28,29]. Since these markers are also expressed on other immune cells, such as neutrophils (e.g., CD15, CD66b), evaluation of direct immunosuppressive function is mandatory for the definition of MDSCs [28].

The immunosuppressive functions of MDSCs are mediated by several secreted factors, including prostaglandin E2 (PGE2), IL-10, transforming growth factor (TGF)-β, nitric oxide (NO) and arginase 1 (Arg1) for Mo-MDSCs [28,30], and reactive oxygen species (ROS), granulocyte-colony stimulating factor (G-CSF) and Arg1 for G-MDSCs [28]. Since Mo-MDSCs are generated from monocytes, and further differentiate to TAMs, Mo-MDSCs and TAMs in human tumors share several cell surface markers [28,30,31]. On the other hand, although several reports have suggested that G-MDSCs are generated from the neutrophil linage, the differentiation of G-MDSCs remains under discussion [28,32]. Notably, both Mo-MDSCs and G-MDSCs correlate with poor prognosis IN cancer patients [32,33]. Targeting MDSCs for the treatment of cancer patients is thus considered to resemble targeting TAMs.

#### 2.2.2. MDSCs and ICIs

Recent reports have suggested the significance of MDSCs in patients with advanced cancer treated using immune checkpoint inhibitors (ICIs) [29]. Increased microRNAs in the plasma of melanoma patients are associated with the generation of MDSCs mediated by melanoma extracellular vesicles, and are even associated with resistance to treatment with ICIs in melanoma patients [29], suggesting that MDSC-related miRs could offer a biomarker of poor prognosis in melanoma patients treated with ICIs. Moreover, among the miRs, a recent report also suggested that miR-150-5p mediates angiogenesis function through the secretion of vascular endothelial growth factor (VEGF) and matrix metalloproteinase (MMP)9 [30]. In another study, hypoxia induced miR-210 to modulate MDSC function by increasing Arg activity and NO production, without affecting ROS, IL6, or IL10 production or expression of PD-L1 [34]. Notably, as we described above, since MDSCs (like TAMs) secrete MMP9 [35] to facilitate MMP9-dependent cleavage of PD-L1 surface expression anti-PD1 Ab resistance [36], hypoxia hinders the anti-tumor effects of anti-PD1 Abs. Since hypoxia-inducible factor (HIF)-1a is one of the key regulators for the differentiation and accumulation of MDSCs in hypoxic tumor regions [37,38], targeting HIF-1a might improve anti-tumor immune responses in patients with anti-PD1 Abs.

#### 2.2.3. Cross-Talk between MDSCs and Other Immunosuppressive Cells

Not only direct immune suppression, MDSCs induce other immunosuppressive cells, such as regulatory T cells (Tregs) and TAMs to maintain the immunosuppressive tumor microenvironment [32]. For example, Hwang et al. reported that Gr1^+^CD115^+^ MDSCs can induce de novo generation of Tregs from adoptively transferred antigen-specific CD25^−^CD4^+^ T cells in the presence of IL-10 and interferon (IFN)-γ in vivo [13]. In another report, MDSCs expanded tumor-specific Tregs via Arg-dependent and TGF-b-independent pathways [39]. On the other hand, Tregs regulated the immunosuppressive function of MDSCs through B7 homologs (B7-H1, B7-H3, B7-H4) in a mouse ret tumor model in vivo [3]. In addition to Tregs, TAMs could also affect MDSC recruitment at the tumor site [40,41,42]. Since several types of MDSCs express CXCR2 [41,42], intratumor production of CXCL5 and CXCL8 is important to migration of MDSCs in the tumor microenvironment [40]. Since one of the main sources of CXCL5 in advanced melanoma is TAMs [35], and CXCL5 could be a predictive biomarker for the efficacy of anti-PD1 Abs in advanced melanoma patients [43], the CXCR2/CXCL5 axis should play a significant role in recruiting MDSCs to the tumor site, and blockade of CXCR2 enhanced anti-tumor immune responses in a melanoma model [44]. Notably, TAMs also produce CCL17 and CCL22 to promote migration of CCR4^+^ Tregs to the tumor site [29]. Since TAMs are a heterogeneous population of cells, and could re-polarize from immunosuppressive M2 phenotypes to classically activated phenotypes by immunotherapy such as type 1 IFN [45] and anti-PD1 Abs [4], these reagents could inhibit migration of CCR2^+^ MDSCs and CCR4^+^ Tregs to the tumor site to induce anti-tumor immune responses in the tumor-bearing host.

In summary, another type of immature macrophage, the MDSC, maintains an immunosuppressive microenvironment by suppressing tumor-specific T cells directly or indirectly. Notably, MDSCs expressed PD-L1, and thus could also represent a target for immunotherapy using anti-PD1 Abs.

### 2.3. Regulatory T Cells: Tregs

#### 2.3.1. Significance of Tregs in Developing Skin Cancers

As described above, Tregs maintain an immunosuppressive tumor microenvironment in skin cancers together with other immunosuppressive cells. Previous report has suggested that a large number of effector (e)Tregs (CD45RA^−^Foxp3^high^CD25^high^) infiltrate tumor sites to induce tolerance by various pathways and thus suppress the function of tumor-specific T cells, contributing to poor prognosis in cancer patients [5]. Notably, eTregs highly express various immune checkpoints, including CTLA4 and PD1, to suppress activated cytotoxic T cells, suggesting that eTregs could represent an optimal target for ICIs such as ipilimumab and nivolumab [5,43,46]. Indeed, Romano et al. reported that ipilimumab depletes CTLA4^+^ Tregs through antigen-dependent cell-mediated cytotoxicity (ADCC) in melanoma patients [46]. In addition, eTregs express inducible T-cell costimulator (ICOS), which promotes the proliferation of activated eTregs by ICOS ligand expressed by plasmacytoid dendritic cells (DCs) [47].

#### 2.3.2. Tregs and ICIs: Anti-PD1 Abs and Anti-CTLA4 Abs

As we described above, eTregs express various immune checkpoints and suppress the cytotoxic function and proliferation of conventional effector T cells to maintain an immunosuppressive tumor microenvironment [5,43]. Indeed, CD45RA^−^Foxp3^high^CD25^high^ eTregs express CTLA4 as well as PD1, ICOS, GITR, OX-40 and LAG3 [43]. CTLA4 expressing Tregs bind to CD80/86 on DCs to inhibit maturation of DCs [48]. Moreover, eTregs produce inhibitory cytokines (TGF-b, IL-10, IL-35) to promote B lymphocyte-induced maturation protein (BLIMP1)-dependent exhaustion of CD8^+^ TILs in the tumor microenvironments of B16 melanoma and the BrafPten melanoma model [49]. In addition to being a therapeutic target, PD1^+^ Tregs are also a useful diagnostic target for anti-PD1 Ab monotherapy [50,51,52]. For example, decreased circulating PD1^+^ Tregs could offer a predictive marker for favorable clinical outcomes from anti-PD1 Abs in advanced melanoma [50]. Moreover, in another report, nivolumab monotherapy in an adjuvant setting decreased circulating PD1^+^ Tregs in stage III melanoma patients [51].

Although nivolumab plus ipilimumab combined therapy is one of the first-line therapies for unresectable melanoma, and is a most effective protocol for BRAF wild-type melanoma, the frequency of serious adverse events is higher than that with anti-PD1 Ab monotherapy [53]. As mentioned above, since ipilimumab depletes CTLA4^+^ Tregs through ADCC, as one of the mechanisms for inducing anti-tumor immune response in melanoma patients that leads to induction of high therapeutic efficacy when administered with nivolumab [46,53], investigations for other drugs that selectively deplete eTregs are ongoing.

For these reasons, several recent studies have targeted eTregs to establish novel anti-PD1 Ab-based immunotherapies [6,52,54]. Among those, Doi et al. reported a phase 1 study of mogamulizumab, an anti-CCR4 Ab, in combination with nivolumab for the treatment of solid tumors [54]. They concluded that mogamulizumab decreased the population of eTregs (CD4^+^CD45RA^−^Foxp3^high^) during treatment, with an acceptable safety profile in combination with nivolumab [49]. More recently, Schoonderwoerd et al. reported that Abs for endothelin, a coreceptor for TGF-β ligands, significantly decreased the number of intratumoral Tregs, leading to enhanced anti-tumor immune response with anti-PD1 Ab therapy [52]. Hu-Lieskovan et al. reported that dabrafenib monotherapy increased TAMs and Tregs in melanoma, which decreased with the addition of trametinib, suggesting that dabrafenib plus trametinib combination therapy could decrease immunosuppressive Tregs, and enhance the anti-tumor effects of anti-PD1 Abs in melanoma patients [6].

Taken together, Tregs suppress tumor-specific T cells, leading to induction of tolerance in the tumor microenvironment in skin cancers. Since Tregs express both PD1 and CTLA4, Tregs could represent an optimal target for nivolumab plus ipilimumab combination therapy.

### 2.4. TANs in Developing Skin Cancers

Neutrophils are polymorphonuclear cells that are classically known to play roles in acute immune responses (e.g., host defense, immune modulation, tissue injury) as one of the innate immune cells [7]. Since oncologists started focusing on cancer inflammation as one of the main facilitators for development of the tumor microenvironment, TANs have recently been taken into accounts as immunosuppressive cells, even in skin cancer [55,56,57]. Indeed, TANs could drive tumor progression through various pathways. For example, TANs not only eliminate the pathogen by phagocytosis, but also lead to DNA base damage and mutation, and subsequent initiation of tumor development [58]. In addition, TANs produce various tumor-driving cytokines such as TGF-β into the tumor microenvironment to maintain macrophages as an M2-polarized phenotype [59], leading to promotion of tumor progression. TANs also produce inducible nitric oxide synthase (iNOS) to directly suppress CD8^+^ effecter T cells at the tumor site [60]. Such reports suggest the significance of inhibiting TAN recruitment at tumor sites.

Among the inducers of TANs, IL-17 could play a significant role in developing skin cancers. Indeed, several reports have suggested the significance of IL-17 in the development of skin cancers such as cutaneous squamous cell carcinoma (cSCC) [61,62] and extramammary Paget’s disease (EMPD) [23]. For example, Wu et al. reported that IL-17 signaling in keratinocytes drives IL-17-dependent sustained activation of the TRAF4-ERK5 axis, leading to keratinocyte proliferation and tumor formation in cSCC [61]. Gasparoto et al. reported a significant correlation between IL-17 and development of mouse cSCC [62]. More recently, a possible correlation of CCL20/IL-23/IL-17 axis in the development of EMPD has been reported [23]. These reports suggest the significance of IL-17 in the carcinogenesis of skin cancers, and IL-17 might be partially caused by the induction of TANs at the tumor site.

In aggregate, TANs are induced by IL-17-related cancer inflammatory factors. TANs produce iNOS to directly suppress the proliferation of effector T cells at a tumor site to promote cancer development.

## 3. Immunosuppressive Myeloid Cells as a Target of Immunotherapy for Skin Cancer

### 3.1. TAMs as a Target for Immunotherapy

Since TAMs could be re-programmed to induce anti-tumor responses, infiltration of CD8^+^ T cells and the presence of TAMs in the tumor microenvironment is mandatory for successful immunotherapy [1,63,64,65]. TAMs promote an immunosuppressive microenvironment together with other immunosuppressive cells by various pathways, and could therefore represent a target for immunotherapy to enhance anti-tumor immune response [1]. Indeed, several recent preclinical reports have suggested significant roles of TAMs in the induction of anti-tumor immune response. For example, leukemia inhibitory factor (LIF) promotes the infiltration of CD163^+^CD206^+^ M2 macrophages, and the blockade of LIF in LIF-expressing tumors increases the production of CXCL9, attracting cytotoxic CD8^+^ T cells to the tumor site [64]. Moreover, since the high number of CD8^+^ T cells at the tumor site is important for immunotherapy, especially using anti-PD1 Abs, LIF-neutralizing Abs in combination with anti-PD1Abs suppress tumor growth and prolong the survival of tumor-bearing hosts [64]. In another report, intratumoral injection of IFN-b also increased CD8^+^ TILs in melanoma by the induction of M1 macrophage-related chemokines (CXCL9, CXCL10, CXCL11), leading to enhancement of the anti-tumor effects of anti-PD1 Abs for melanoma in vivo [17,66]. Those reports suggest that induction of CD8^+^ cytotoxic T cells at the tumor site by re-polarizing TAMs to produce M1-related chemokines could enhance the anti-tumor immune response to anti-PD1 Abs. Notably, a recent report suggested that not only M1/M2 TAMs, but also TAMs that produce proinflammatory chemokines, should be taken into account in the development of cutaneous melanoma [45]. These subsets of TAMs might be induced by differences in the conditioning of newly recruited monocytes or a shift in the proportion of PD1^−^/PD1^+^ TAM subsets [67]. These TAM subsets might represent another target for anti-PD1 Abs. In future, further clinical trials are mandatory before these drugs can be applied to clinical use [67].

Re-polarization from a non-inflammatory phenotype toward an inflammatory phenotype of TAMs is important for the treatment of melanoma, which shows high glycolytic activity, leading to increased acidosis in the tumor microenvironment [63]. They demonstrated that inhibition of tumor acidosis by adenylyl cyclase inhibitor MDL-12 induces NOS, CXCL9, CXCL11 and TNF-α-expressing proinflammatory TAMs that phenotypically resemble classical M1 macrophages, leading to suppression of tumor growth in B16 mouse melanoma [63]. In another report, Zhang et al. reported that infusions of nanoparticles formulated with mRNAs encoding IFN regulatory factor 5 (IRF5) in combination with IKKβ re-programs the immunosuppressive TAMs toward anti-tumor immunity, leading to induction of tumor regression [68]. These reports suggested that TAMs could be one of the optimal targets for the development of novel immunotherapies in future.

### 3.2. MDSCs as a Target for Immunotherapy

Since MDSCs are one of the key immunosuppressive cells involved in maintaining the tumor microenvironment in each skin cancers, abrogation of the suppressive function of MDSCs delays tumor growth in skin cancers. For example, Nam et al. reported the significance of IFN regulatory factor (IRF)4 for the functional differentiation of MDSCs by myeloid-specific deletion of IRF4 to abrogate the inhibitory effects of MDSCs in suppressing T-cell proliferation [69]. More recently, Pan et al. reported that increased expression of ten-eleven translocation-2 (tet2) on tumor-infiltrating myeloid cells maintained the suppressive function of immature myeloid cells through IL-1R/MyD88 pathways in melanoma patients [70]. Sierra et al. reported the therapeutic effect of the humanized anti-Jagged1/2-blocking antibody CTX014 in the treatment of mouse B16F10 melanoma [71]. Since CTX014 inhibits the accumulation of MDSCs as well as the expression of Arg1 and iNOS, both of which are known to suppress T-cell proliferation, on MDSCs, CTX014 induces melanoma-specific T cells to suppress tumor growth in the melanoma-bearing host [71]. In aggregate, these reports suggest that functional modulation of MDSCs could improve the anti-tumor immune response in tumor-bearing hosts.

Since MDSCs directly suppress tumor-specific T cells in the tumor-bearing host, suppression of MDSC accumulation at the tumor site requires establishment of the appropriate immunotherapy. In addition to CXCR2 [41,42], MDSCs could increase the expression of CCR5, which is reported to present highly immunosuppressive phenotypes of MDSCs, by stimulating IL-6 and granulocyte macrophage-colony stimulating factor (GM-SCF) [72]. Moreover, CCR5^+^ MDSCs in melanoma patients correlate with enhanced production of CCR5 ligands, facilitating the accumulation of CCR5^+^ MDSCs at the tumor site [72]. Selective blockade of CCR5 could thus reduce migration of the subpopulation of MDSCs that possesses immunosuppressive potential, leading to improved anti-tumor immune response in the tumor-bearing host [72]. In another report, Shi et al. reported that CXCL1 and CXCL2 derived from B16F10 melanoma cells promoted the generation of Mo-MDSC in the bone marrow of melanoma-bearing mice, but did not correlate with the chemotaxis and proliferation of MoMDSCs at the tumor site [73]. They concluded that blockade of CXCL1 and CXCL2 could improve immune tolerance at the tumor site by decreasing MoMDSCs in melanoma [73].

In summary, both TAMs and MDSCs produce characteristic chemokines by the stimulation of stromal factors in each cancer species. Since several drugs can modulate the production of these chemokines, immunosuppressive myeloid cells could be an optimal target for immunotherapy in skin cancers.

## 4. TAM-Related Biomarkers for Predicting the Efficacy of ICIs

Recently, biomarkers for predicting efficacy and immune-related adverse events (irAEs) for anti-PD1 Abs have been widely investigated [67]. For example, routine blood tests such as neutrophil-to-lymphocyte ratio (NLR) and serum lactate dehydrogenase (LDH) are clinically used to predict the efficacy of anti-PD1 Abs [74,75,76,77]. PD-L1 expression on melanoma cells has been reported as an independent prognostic factor that correlates with vertical invasion of melanoma cells [78]. PD-L1 is also expressed on TAMs in various cancer species [74,79]. Among these, PD-L1 expression on TAMs could offer a prognostic biomarker for esophageal carcinoma [79], suggesting that expression of PD-L1 on TAMs might provide a predictive biomarker for anti-PD1 Ab therapy.

As described above, TAMs in melanoma patients express not only PD-L1, but also PD-1 [4]. Because PD-1 expression in TAMs is one of the key factors in M2 macrophage polarization [4], administration of an anti-PD1 antibody might repolarize TAMs, leading to TAM activation and supporting anti-tumor immune response by the production of various chemokines as described in Section 3. Since the main population of TAMs in skin cancer is CD163^+^ M2 macrophages, TAM activation releases soluble (s)CD163, suggesting its utility as a prognostic marker for anti-PD1 antibody treatment. Indeed, increased serum levels of sCD163 correlated significantly with the efficacy of nivolumab for cutaneous melanoma (84.6% sensitivity, 87.0% specificity; *p* = 0.0030) [80], and development of irAEs caused by nivolumab (*p* = 0.0018) [14]. Those reports suggested that TAM-related sCD163 could provide a predictive marker for the efficacy and irAEs of anti-PD1 Abs.

Not only sCD163, but also TAM-related chemokines could provide prognostic markers for the efficacy or development of irAEs by anti-PD1 Ab treatment [81,82]. Baseline serum CXCL5, but not CXCL10 and CCL22, is associated with the efficacy of nivolumab in advanced melanoma [81]. Moreover, increased absolute serum levels of CXCL5 correlated significantly with irAEs by nivolumab [14]. Increased serum levels of CCL19 also correlate significantly with outcomes of anti-PD1 Abs and development of vitiligo in advanced melanoma patients [82]. These reports suggested that TAM-related chemokines could provide prognostic markers for anti-PD1 Ab therapy in advanced melanoma.

Biomarkers for predicting efficacy and irAEs for anti-PD1 Abs have been widely investigated, but little is known. Among these, TAM-related factors might offer optimal markers to predict the efficacy of anti-PD1 Ab monotherapy.

## 5. Roles of TAMs in Immune-Related Adverse Events irAEs

Since ICIs have been widely used against various cancer species for the past 5 years, series of irAEs have also been reported [83]. Among these, Darnell et al. reviewed and classified irAEs caused by ICI, suggesting that especially in systemic organs, these irAEs resemble conventional autoimmune diseases [83]. For example, anti-PD1 Abs could induce pathologies such as lymphatic colitis, hyperthyroidism (Grave’s disease), isolated ACTH deficiency [84], Vogt-Koyanagi Harada disease-like uveitis [85], pneumonitis, and autoimmune-like skin diseases, all of which are similar to the conventional courses of these diseases. Since anti-PD1 Abs also induce autoimmune like skin diseases such as bullous pemphigoid (BP) [86], psoriasis [87], vitiligo [82], lichen planus [88] and alopecia areata [89], the investigation of cutaneous microenvironments in such conventional skin diseases is useful for understanding the mechanisms involved in the induction of irAEs.

Among the cutaneous disorders described above, focus has recently been placed on the immunomodulatory roles of CD163^+^ tissue-resident macrophages to understand the possible mechanisms of BP [90,91,92] and psoriasis [93]. For example, substantial numbers of CD163^+^CD206^+^ M2 macrophages are detected in the lesional skin of BP [90]. These CD163^+^ macrophages produce sCD163, Th2 type chemokines such as CCL17 and CCL22 and MMP9 [91,92], leading to the development of a Th2-polarized immune microenvironment in the lesional skin or blister fluid of BP. On the other hand, the lesional skin of psoriasis possesses heterogeneous CD163^+^ M2-polarized macrophages, including MARCO^+^ phenotypes of M2 macrophages, as well as IL-23- or TNFa-producing macrophages that contribute to the Th17-polarized inflammatory microenvironment of psoriasis [93]. In aggregate, these reports suggest that CD163^+^ M2-polarized macrophages could contribute to the pathogenesis of autoimmune skin diseases such as BP and psoriasis.

As described above, CD163^+^ TAMs in melanoma patients express not only PD-L1, but also PD-1 [4]. Since TAMs express PD1, which is necessary for maintaining M2 phenotypes of CD163^+^ TAMs by PD-L1/PD1 signaling [4], and since the blockade of PD-L1/PD1 re-polarizes and activates TAMs into antitumor, anti-PD1 Abs could activate CD163^+^ TAMs when these Abs reach at the tumor microenvironment appropriately (Figure 1). Notably, the main population of TAMs in skin cancer is CD163^+^ M2 macrophages [1], with sCD163 as the activation marker [94], suggesting that CD163 activated with PD1 antibody should release sCD163 in the tumor microenvironment. Indeed, sCD163 has been reported as a biomarker for predicting both the efficacy of anti-PD1 Abs [80] and irAEs caused by nivolumab [14]. In addition, Freeman-Keller reported an epidemiological correlation between the efficacy of nivolumab and irAEs caused by this agent [95]. Several retrospective studies and case reports have also supported this hypothesis [95,96,97,98]. Taken together, these reports suggested correlations between the efficacy of anti-PD1 Abs and several irAEs, and that CD163^+^ TAMs play a key role in inducing the efficacy of anti-PD1 Abs as well as several irAEs.

In summary, since TAMs express PD1 and PD-L1, the irAEs caused by anti-PD1 Abs might be, at least in part, caused by the blockade of PD1/PD-L1 signaling on TAMs. Further studies are needed to elucidate the mechanisms underlying irAEs by ICIs in the future.

## 6. Concluding Remarks

Since TAMs express PD1 to maintain an immunosuppressive M2 phenotype by PD1/PD-L1 signaling from tumor cells, and blockade of PD1/PD-L1 signaling by anti-PD1 Abs activates and re-polarizes TAMs toward an immunoreactive M1 phenotype, TAMs could represent a useful target for anti-PD1 Abs. Moreover, not only TAMs, but also MDSCs express PD-L1, and Tregs express both PD1 and CTLA4. These immunosuppressive cells could therefore also be targets for immunotherapy using anti-PD1 Abs. Since several preclinical studies have reported enhanced effects from anti-PD1 Ab therapy targeting TAMs, TAM-targeting therapies for advanced melanoma and non-melanoma skin cancer will develop in the future.

## Figures and Tables

**Figure 1 biomolecules-10-01087-f001:**
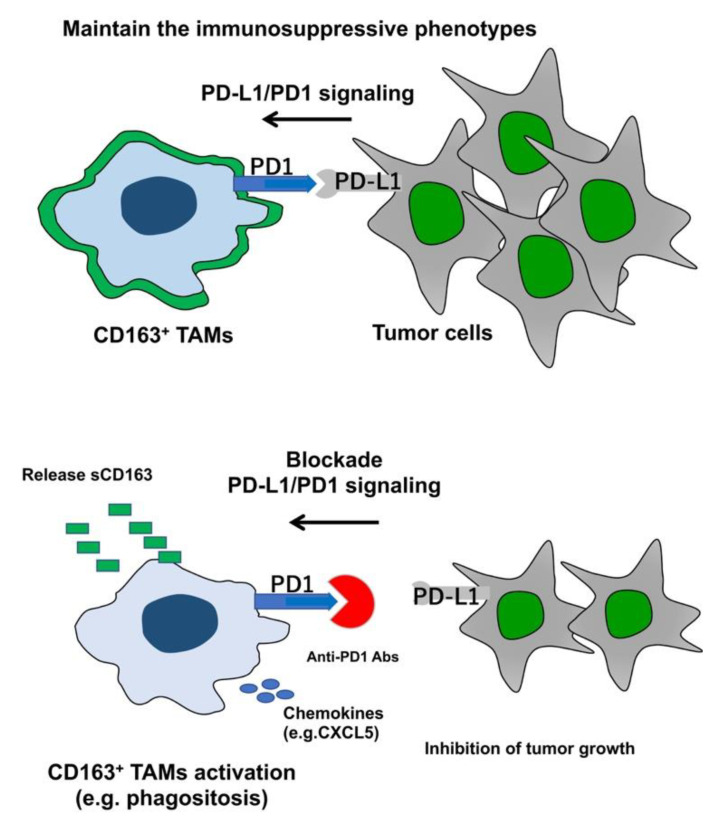
Schematic representation of TAMs during anti-PD Abs monotherapy. TAMs in melanoma maintain immunosuppressive function through PD1/PD-L1 signals. Blockade of PD1/PD-L1 signaling activates CD163^+^ TAMs to inhibit tumor growth.

**Table 1 biomolecules-10-01087-t001:** Positive and negative markers for TAMs and MDSCs.

	Subtypes	Positive	Negative
TAMs	M1	CD68, CD86, CD169, HLA-DR, CCR7	
	M2	CD163, CD204, CD206, PD-L1, ARG1	
MDSCs	MoMDSC	CD11b, PGE2, IL-10, TGFb, iNOS, ARG1	HLA-DR, CD14
	G-MDSC	CD15, CD33, CD66b, ROS, G-CSF, ARG1	HLA-DR, CD14

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
