# Peer review of "Significance of Immunosuppressive Cells as a Target for Immunotherapies in Melanoma and Non-Melanoma Skin Cancers"

_biomolecules, 2020, doi:10.3390/biom10081087_

Round 1
Reviewer 1 Report
Immunosuppression is a major factor for the development and sustained growth of malignant cells. Macrophages play an essential role in clearing pathogens and help in eliminating tumors. However, establihed tumor cells manipulate the immune cells, including macrophages, for their survival and growth. Recent studies have established the presence of M1 and M2 macrophages. This review article summarizes the key differences and requirements of these two macrophage subsets. In particular, the expression and functions of checkpoint molecues such as PD1 and its ligand PD-L1. This review article is overall written well and contains key details. However, the following concerns should be addressed by the authors before it can be reviewed further.
- There are multiple long-winded circular arguements starting from the abstract (lines 18-21; 21-24).
- There are also discontinuity in the way information is presented (Introduction). Also, this section is too short to introduce the subject area.
- It is strongly suggested that the authors rewrite the Introduction section to capture the salient aspects of the immunosuppressive cells discussed in the review. An overarching concept and hypothesis should be developed in the Introduction.
- The aspect of 'skin cancer' focus is not well-articulated in the abstract or Introduction. This needs further work in these sections.
- Each individual section do not end with a take-home message. Some level of summarizing (at least two sentences) following each section can help the reader.
- Figures need to be vastly improved. They absolutely do not capture the information presented in the text.
- Authors are strongly encouraged to add additional figures to help the readers with the details they present in the text.
Author Response
Reviewers' comments:
Reviewer: 1
Comments to the Author
Immunosuppression is a major factor for the development and sustained growth of malignant cells. Macrophages play an essential role in clearing pathogens and help in eliminating tumors. However, established tumor cells manipulate the immune cells, including macrophages, for their survival and growth. Recent studies have established the presence of M1 and M2 macrophages. This review article summarizes the key differences and requirements of these two macrophage subsets. In particular, the expression and functions of checkpoint molecues such as PD1 and its ligand PD-L1. This review article is overall written well and contains key details. However, the following concerns should be addressed by the authors before it can be reviewed further.
1. There are multiple long-winded circular arguements starting from the abstract (lines 18-21; 21-24).
Thank you for this comment. We have amended the sentences identified.
2. There are also discontinuity in the way information is presented (Introduction). Also, this section is too short to introduce the subject area.
We have rewritten the Introduction section to address these concerns.
3. It is strongly suggested that the authors rewrite the Introduction section to capture the salient aspects of the immunosuppressive cells discussed in the review. An overarching concept and hypothesis should be developed in the Introduction.
Thank you for your comments. We have rewritten the Introduction section based on the concerns raised by you and Reviewer 2.
4. The aspect of 'skin cancer' focus is not well-articulated in the abstract or Introduction. This needs further work in these sections.
Thank you for raising this issue. As mentioned in the Introduction section, environmental risk factors for skin cancer (e.g., sun exposure, chemical exposure) have been widely reported. These risk factors modulate the profiles of tumor-infiltrating leukocytes (TILs) through AhR-dependent signal pathways, and chronic exposure to AhR ligands at skin lesion sites induce chronic inflammation including macrophages, neutrophils and T cells. We therefore consider skin cancer as one of the optimal models to discuss the development of immunosuppressive microenvironment in cancers. We have added sentences to address these issues in the Introduction section.
5. Each individual section do not end with a take-home message. Some level of summarizing (at least two sentences) following each section can help the reader.
Thank you for this comment. We have added summary sentences as pointed out.
6. Figures need to be vastly improved. They absolutely do not capture the information presented in the text.
We have changed the figures as suggested.
7. Authors are strongly encouraged to add additional figures to help the readers with the details they present in the text.
We have added additional figures in accordance with this comment.
Reviewer 2 Report
This review compiles most of the more relevant advances achieved during the last 5 years related with the issue defined at the title of the manuscript.
Several things to be corrected and or discussed in the text:
- I think authors are underestimating their concluding remark. PD1/PDL1 blockade is a promising immune-therapeutical approach precisely because it affects not only to TAMs, but also simultaneously to additional key cells from both the innate and the adaptive arms of the immune system, as the own authors compile along the manuscript (MDSCs, TIL and Treg). This point should be therefore highlighted.
- This review is mainly based on the seminal paper by Gordon et al. (Nature 2017) who first identified in murine models the potential role of TAM PD1 in tumor progression. The existence of PD1+ TAM subsets has been later reinforced in additional murine models (gastric cancer and squamous cell carcinoma) by two additional papers, which should be commented and included at the reference list (Furong Wang et al. Oncogenesis 2018, 7:41; Bin Li et al. Oncogenesis 2019, 8:17).
- I am not sure if it has been really demonstrated that reprogramming of TAMs from M2>M1 phenotype is the (only) mechanism explaining the effect of PD1 blockade in vivo, since differentiated macrophages are not that easy to be reprogrammed (Ramón R Rodríguez et al. Cell Reports 2019, 29:860). Maybe a different conditioning of newly recruited monocytes and/or a shift in the proportion of PD1-/PD1+ TAM subsets should also be considered as mechanistic pathways for PD1 blockade effects. These, or alternative ideas, should be shortly discussed in the text. Indeed, the M1-M2 classification of TAMs is not totally apropriated in cancer. As the own authors say in the abstract: ‘the main population of TAMs comprises CD163+ M2 macrophages… which release soluble pro-inflammatory chemokines’. Although conceptually this is somehow contradictive, it seems to be the real scenario since TAMs have characteristics of both M1 (pro-inflammatory mediators) and M2 (immunosuppressive) macrophages, see for instance Samaniego et al. Cancer Immunology Research, 2018 for human melanoma TAMs, or Antsiferova et al. EMBO Mol Med 2016 for TAMs at HPV8-skin murine model.
- Common and different markers between human TAMs and MDSCs should be specified (around line 112), as this would be informative. Indeed, is there any paper demonstrating the existence or comparing the relative abundance of each cell type in human skin cancer?
- References 11 and 65, and 3 and 39, are duplicated.
Author Response
This review compiles most of the more relevant advances achieved during the last 5 years related with the issue defined at the title of the manuscript.
Several things to be corrected and or discussed in the text:
1. I think authors are underestimating their concluding remark. PD1/PDL1 blockade is a promising immune-therapeutical approach precisely because it affects not only to TAMs, but also simultaneously to additional key cells from both the innate and the adaptive arms of the immune system, as the own authors compile along the manuscript (MDSCs, TIL and Treg). This point should be therefore highlighted.
Thank you for your comment. We have added the sentences as you pointed out.
2. This review is mainly based on the seminal paper by Gordon et al. (Nature 2017) who first identified in murine models the potential role of TAM PD1 in tumor progression. The existence of PD1+ TAM subsets has been later reinforced in additional murine models (gastric cancer and squamous cell carcinoma) by two additional papers, which should be commented and included at the reference list (Furong Wang et al. Oncogenesis 2018, 7:41; Bin Li et al. Oncogenesis 2019, 8:17).
Thank you for this comment. We have added the references you suggested.
3. I am not sure if it has been really demonstrated that reprogramming of TAMs from M2>M1 phenotype is the (only) mechanism explaining the effect of PD1 blockade in vivo, since differentiated macrophages are not that easy to be reprogrammed (Ramón R Rodríguez et al. Cell Reports 2019, 29:860). Maybe a different conditioning of newly recruited monocytes and/or a shift in the proportion of PD1-/PD1+ TAM subsets should also be considered as mechanistic pathways for PD1 blockade effects. These, or alternative ideas, should be shortly discussed in the text. Indeed, the M1-M2 classification of TAMs is not totally apropriated in cancer. As the own authors say in the abstract: ‘the main population of TAMs comprises CD163+ M2 macrophages… which release soluble pro-inflammatory chemokines’. Although conceptually this is somehow contradictive, it seems to be the real scenario since TAMs have characteristics of both M1 (pro-inflammatory mediators) and M2 (immunosuppressive) macrophages, see for instance Samaniego et al. Cancer Immunology Research, 2018 for human melanoma TAMs, or Antsiferova et al. EMBO Mol Med 2016 for TAMs at HPV8-skin murine model.
Thank you for this pertinent comment. We have added the sentences and references you suggested.
4. Common and different markers between human TAMs and MDSCs should be specified(around line 112), as this would be informative. Indeed, is there any paper demonstrating the existence or comparing the relative abundance of each cell type in human skin cancer?
We have added this information to Table 1, as suggested.
5. References 11 and 65, and 3 and 39, are duplicated.
Thank you for noticing these errors. We have deleted the duplications and have checked the references.